# A Review of Endogenous and Exogenous Contrast Agents Used in Photoacoustic Tomography with Different Sensing Configurations

**DOI:** 10.3390/s20195595

**Published:** 2020-09-29

**Authors:** Victor T.C. Tsang, Xiufeng Li, Terence T.W. Wong

**Affiliations:** Translational and Advanced Bioimaging Laboratory, Department of Chemical and Biological Engineering, Hong Kong University of Science and Technology, Hong Kong, China; tcvtsang@connect.ust.hk (V.T.C.T.); xlies@connect.ust.hk (X.L.)

**Keywords:** photoacoustic tomography, label-free imaging, exogenous contrast agents, multispectral imaging, all-optical detection

## Abstract

Optical-based sensing approaches have long been an indispensable way to detect molecules in biological tissues for various biomedical research and applications. The advancement in optical microscopy is one of the main drivers for discoveries and innovations in both life science and biomedical imaging. However, the shallow imaging depth due to the use of ballistic photons fundamentally limits optical imaging approaches’ translational potential to a clinical setting. Photoacoustic (PA) tomography (PAT) is a rapidly growing hybrid imaging modality that is capable of acoustically detecting optical contrast. PAT uniquely enjoys high-resolution deep-tissue imaging owing to the utilization of diffused photons. The exploration of endogenous contrast agents and the development of exogenous contrast agents further improve the molecular specificity for PAT. PAT’s versatile design and non-invasive nature have proven its great potential as a biomedical imaging tool for a multitude of biomedical applications. In this review, representative endogenous and exogenous PA contrast agents will be introduced alongside common PAT system configurations, including the latest advances of all-optical acoustic sensing techniques.

## 1. Introduction

Probing the molecular contrast is one of the key advantages of optical-based sensing approaches. By adding the spatial dimension, the development of optical microscopy—from classical bright-field microscopy to other advanced optical imaging modalities (e.g., multiphoton and light-sheet microscopy), has been one of the main drivers for discoveries and researches in both life science and biomedical imaging. These optical imaging modalities can provide rich molecular contrast and generate high-resolution images for visualizing subcellular features such as organelles. However, pure optical imaging modalities utilize the photons in the ballistic regime only, leading to an imaging depth limit ~1 mm due to strong optical scattering in biological tissues. The shallow imaging depth intrinsically limits optical imaging approaches’ translational potential to a clinical setting [1,2,3]. Therefore, there is a huge demand for biomedical imaging modalities that can provide non-invasive deep-tissue imaging with high spatial resolution. Biomedical imaging modalities such as magnetic resonance imaging (MRI), X-ray computed tomography (CT), positron emission tomography (PET), and ultrasound (US) imaging are commonly used in clinical settings owing to their deep imaging depth and adequate resolution for diagnosis. However, PET and CT utilize ionizing radiations, which might not be suitable for longitudinal studies that require frequent and prolonged exposure. Instead, US and MRI are non-invasive deep-tissue imaging modalities that can be used in numerous situations when safety is the main consideration. However, due to the low molecular specificity, contrast agents such as microbubbles and gadolinium-based contrast agents are used to improve the molecular contrast and reveal metabolic information. Nevertheless, recent studies have shown that heavy metals atoms such as the gadolinium used for MRI contrast accumulate in human bodies [4]. Nowadays, there is not enough clinical evidence to show any significant side effects for gadolinium bioaccumulation. However, biocompatible and efficient contrast agents are the key directions for achieving safe and specific molecular sensing in biomedical imaging.

Photoacoustic tomography (PAT) is a rapidly growing hybrid imaging modality that converts light energy to ultrasonic waves through the photoacoustic (PA) effect. Typically, when laser pulses are illuminated onto the targeted biological tissue, some photons will be locally absorbed and converted into heat, leading to transient thermoelastic expansion, and hence generating wideband PA waves. The PA waves are subsequently detected by a single-element or an array-based ultrasonic transducer (UT), and reconstructed into images. The amplitude of the PA signal is proportional to the optical absorption coefficient of the biomolecule, providing 100% sensitivity to variation in optical absorption contrast. As the attenuation of PA waves in biological tissues is ~3 orders of magnitude weaker than that of photons, the ultrasonic detection nature of PAT allows deeper imaging depth at high resolution when compared with optical microscopy. To achieve deep-tissue imaging, near-infrared (NIR) light is highly preferred as it provides a good balance between optical scattering and absorption. NIR-I (700–950 nm) enables deep tissue penetration in muscle, stomach, heart, and brain tissue, while NIR-II (1000–1700 nm) shows deeper penetration depth when imaging the kidney, spleen, and liver [5]. PAT has two major modes according to the imaging reconstruction strategy. The first one is photoacoustic microscopy (PAM), where a single focused UT is usually used for sensitive detection to form an image. Based on the resulting imaging resolution, PAM can be further divided into optical-resolution PAM (OR-PAM) and acoustic-resolution PAM (AR-PAM). For the other mode, photoacoustic computed tomography (PACT), PA signals are usually excited by a wide-field laser illumination. The PA signals can be detected through an array-based UT, and reconstructed into images using back-projection algorithms. With the versatility of PAT, researchers have proven its great potential as a biomedical imaging tool for molecular imaging, diagnosis, and imaging-guided therapy [1,3,6,7,8].

In this review, we first introduce how do different endogenous contrasts provide information for biomedical applications through PAT. Then, by understanding the applications and limitations of endogenous contrasts, we will introduce the development of exogenous contrasts and their applications in PAT. Lastly, with the understanding of the information available for PAT with or without exogenous contrasts, common configurations of reflection-mode PAT systems and the latest advances in different all-optical acoustic sensing techniques used in PAT will be highlighted.

## 2. Endogenous Contrast Agents

One of the main advantages of PAT is its label-free imaging capability with high optical absorption contrast. The spectrum of some major molecules in biological tissue is shown in Figure 1. Intrinsic chromophores (e.g., hemoglobin and melanin) can be used to provide structural and functional information for biomedical applications, including cancer prognosis and small animal’s brain activity measurement.

### 2.1. Hemoglobin

Hemoglobin, which is responsible for delivering oxygen via the bloodstream, is an excellent endogenous contrast. Imaging hemoglobin with PAT can help to visualize the development of new vasculatures such as angiogenesis, which is a crucial parameter for early detection of tumor development [10]. Multiple studies had demonstrated the ability of PAT in early-stage cancer detection by monitoring the change in vasculature density [11,12,13]. Lin et al. has demonstrated the use of a PACT system in detecting human breast tumor with a penetration depth of 4 cm within a single breath-hold [14]. Moreover, the non-invasive nature of PAT allows longitudinal studies of morphological changes in the complex vasculature during and after therapies. This non-invasive vascular monitoring has showcased PAT’s potential to be used in many biomedical applications, including the evaluation of the effectiveness of vascular targeting treatments [15], the study of the histopathological processes of early tumor development, and the identification of early cancer prognosis [16].

In addition to providing anatomical information, functional information associated with blood oxygen saturation (SO_2_) can also be extracted in vivo. Heme, an iron-protoporphyrin moiety, is the prosthetic group presents in hemoglobin that allows the binding of oxygen atoms. The ferrous iron atom is coordinated out of plane by four nitrogen atoms and nitrogen atom from the imidazole ring in the proximal histidine. Upon oxygen binding, the ferrous iron changes to a ferric iron atom, with a smaller ion size, and aligns in-plane to the protoporphyrin ring [17,18]. The oxygenation and geometrical changes of the heme group result in differences between the absorption spectra of deoxyhemoglobin and oxyhemoglobin [19]. In a typical PAT system, this quantitative measurement can be obtained by decoupling the PA signals generated by both oxyhemoglobin and deoxyhemoglobin using light sources with two different wavelengths, which is similar to blood oxygen level-dependent (BOLD) signals in fMRI. However, BOLD signals only reflect the magnetic variations caused by deoxyhemoglobin, but not oxyhemoglobin. Together with the high temporal resolution of PAT, it is possible to indirectly measure neuronal activities via the neurovascular coupling. Functional PAT has been demonstrated in small animal models for neuronal studies like brain metabolism and connectome [20,21]. Zhang et al. developed a functional PAM (fPAM) system that can monitor the SO_2_ of subcutaneous vessels in vivo, and achieved a depth-to-resolution ratio of over 100 [22]. Yao et al. demonstrated the use of a high-speed fPAM system for hemodynamic imaging of a mouse brain. In particular, instead of using multiple wavelengths to quantify the amount of oxyhemoglobin and deoxyhemoglobin, the proposed system was able to measure the SO_2_ with a single wavelength by a pulse-width-based method [23], simplifying the system design due to chromatic aberration. In 2019, Liu et al. proposed a nonlinear iteration method to compensate for the absorption saturation effect in SO_2_ measurement in a conventional linear model with three excitation wavelengths, which further improved the accuracy of the quantitative measurements [24].

### 2.2. Deoxyribonucleic/Ribonucleic Acids

Deoxyribonucleic acid (DNA) is a genetic material present in every cell nucleus, which is ubiquitous to every biological tissue. Ribonucleic acid (RNA) is a similar molecule that converts that genetic code into proteins to carry out cellular functions. DNA/RNA has a strong optical absorption at the deep-ultraviolet (UV) wavelength region, naturally being an excellent endogenous contrast for PAT. From previous studies, upon UV laser illumination, cell nuclei are highlighted due to the DNA/RNA contrast, which resembles the contrast used in histopathological images that arise by hematoxylin and eosin (H&E) staining [25]. UV-PAM has demonstrated its ability to generate histology-like images for fixed and unprocessed breast tumor tissues with a lateral resolution down to ~330 nm where individual cell nuclei can be identified. H&E-stained images of the same breast tumor samples were used to validate the images acquired by the UV-PAM system. Histologic features such as ductal carcinoma in situ can sufficiently be identified by a pathologist, affirming the ability for the UV-PAM system to be used in a post-operative setting [25]. To implement UV-PAM as an intra-operative tool, the image acquisition time has to be shortened, which can be achieved by increasing the laser repetition rate and speeding up the scanning mechanism [26]. The DNA/RNA endogenous contrast enables UV-PAM to provide histology-like images without any tissue staining, as shown in Figure 2, which reduces the preparation work that is inevitable in conventional histology. UV-PAM has showcased the potential benefits of translating PAT to a clinical setting.

### 2.3. Melanin

Melanin is a ubiquitous pigment found in skin, hair, and eyes, which is a strong light absorber. Therefore, melanin production is thought to be responsible for protecting human skin from overexposure to harmful UV radiation. Melanin is produced in melanocytes, which might mutate and develop into malignant melanomas. Therefore, melanin, as an endogenous contrast, can be used to reveal melanomas and metastatic melanoma cells [27]. PAT can be used to non-invasively image the subcutaneous melanoma and the surrounding vasculature, providing three-dimensional (3D) information for both melanoma diagnosis and therapy planning. Nevertheless, melanin contrast was also used for PA ophthalmoscopy to study degenerative retinal diseases [28].

### 2.4. Lipid

Lipid is an essential class of molecules that act as biosynthetic precursors and biomolecule for energy storage. Revealing the distribution of lipid can help us understand the initiation and progression of lipid-based diseases such as coronary and peripheral artery diseases. In addition, lipid is a large class of molecules that all have rich hydrocarbon (CH) single bonds in common. Specific lipids, such as myelin in neuronal tissues, can be imaged without label using nonlinear vibrational microscopy, which is based on coherent anti-Stokes Raman scattering (CARS). Although CARS microscopy can offer a submicron resolution to reveal subcellular features [29], it has a shallow imaging depth of <200 μm due to turbidity and heterogeneity of biological tissue, limiting its potential for deep-tissue imaging for clinical purposes. In contrast, PAT can provide sensitive, specific, high-resolution, and deep imaging depth for lipids. Excitation wavelength such as ~1210 and ~1720 nm can be utilized for visualizing lipid in PAT, which can provide positive contrast for lipids from water as background in biological tissues (Figure 3a) [30,31,32].

In 2014, Matthews et al. developed a PAM system to image nerve samples from mice and attained a lateral resolution of 2.7 µm under an excitation wavelength of 1210 nm [32]. The lipid-rich myelin surrounding the peripheral nerve was obviously visualized, and it has a high correlation coefficient with that in the stained image (Figure 3b,c).

To achieve a cost-efficient excitation laser for lipid imaging, Buma et al. developed a multispectral PAM for lipid imaging with a pulsed-supercontinuum laser [33]. The 1225 and 1714 nm wavelengths were used to highlight lipids, while 1325 and 1600 nm wavelengths were selected to provide image contrast for water. They have demonstrated that the PAM image of lipid has a higher contrast at 1714 nm than that at 1225 nm. In addition, a Grüneisen-relaxation PAM system at 1725 nm excitation was proposed to enhance the contrast by a factor of 8.26 [34].

Wang et al. reported a method termed vibrational PA (VPA) microscopy that enables 3D vibrational imaging of tissues with millimeter-scale penetration depth and field-of-view [35]. VPA microscopy can specifically highlight lipid moieties by the excitation of second overtone of CH bond stretching mode, lying in between 1000 and 1300 nm where light is the least absorbed by biological tissues. Since VPA microscopy is based on the excitation of molecular overtone vibrational states, such technique can be applied to other bonding overtone stretching modes such as NH and OH bonds, which are common bonding in many biomolecules. In terms of biomedical application, VPA microscopy had showcased the ability to image lipid-rich atherosclerosis plague from the lumen side in 3D. Later on, the principle of VPA microscopy has been further developed into the intravenous PA (IVPA) catheter, which demonstrated accurate lipid localization and quantification in rabbit aorta in vivo, and in fresh human coronary arteries ex vivo, indicating the translational and clinical potential for intravital diagnosis of lipid-related disorders [35].

### 2.5. Photoacoustic Tomography with Multiple Endogenous Contrasts

3D histological imaging of organs can provide histopathological interpretation from organelles to the organ level, allowing a better understanding of spatial and functional relationships. Conventional 3D imaging modalities often require laborious specimen handling or optical tissue clearing, which suffers from uncertain loss of biological information. However, PAT can rely on abundant intrinsic biomolecules such as hemoglobin, melanin, lipid, and DNA/RNA to provide rich label-free contrast at wavelengths ranging from UV to near-infrared (NIR) regions. Moreover, deep imaging depth achieved by excitation light-limited penetration allows less tissue sectioning, retaining the structural integrity for high fidelity 3D whole-organ imaging. Wong et al. had developed the first automated microtomy-assisted PAM (mPAM) which utilized dual wavelengths (266 and 420 nm) illumination for label-free 3D whole-brain imaging with a lateral resolution of 0.91 μm, revealing individual cell nuclei clearly in transverse, coronal, and sagittal views in the final volumetric image [36]. mPAM system has demonstrated the ability to simultaneously visualize anatomical structures such as cell nuclei, blood vessels, and neuron myelination in the brain, as shown in Figure 4, utilizing multiple endogenous contrasts in PAT where several fluorescence probes/dyes might be required in conventional 3D imaging modalities [36].

In 2017, Diot et al. introduced a handheld multispectral optoacoustic tomography (MSOT) system that aims to identify the spatial distributions of different biomolecules in both breast cancer and non-cancerous breast tissue for early diagnosis, and to monitor the development of the disease and treatment efficacy [13]. MSOT system can simultaneously excite water, lipid, oxyhemoglobin, and deoxyhemoglobin with a fast wavelength scanning laser that illuminates the breast tissue with 28 wavelengths within 700–970 nm. Each absorber’s image is resolved by first filtering the acoustic signals, and later linearly unmixing with respect to the four endogenous contrasts’ absorption spectra within 700–970 nm. The composite image of all four absorbers reveals the anatomical structure of the breast tissue [13]. The same MSOT system has been applied to image thyroid [37]. Similarly, by using multiple wavelengths and also its readily available ultrasound imaging mode, muscle, carotid, and thyroid can be resolved (Figure 5). The second stage clinical trial for MSOT will be needed for the relation of the identified spatial distributions of different biomolecules to clinical diagnostic and therapeutic applications [13,38].

After all, the ability for PAT systems, like mPAM and MSOT, to utilize multiple endogenous contrasts have showcased their potential to be used as an innovative and highly practical biomedical imaging tool to better understand complex biological organs.

## 3. Exogenous Contrast Agents

Biomedical applications of PAT with endogenous contrast had been studied extensively in the past decade. However, endogenous contrast can only give access to limited biological processes because not all biomolecules will have a distinctive optical absorption spectrum. Without an optimized wavelength selection and spectral unmixing, endogenous contrast could easily act as a strong background masking out the molecules of interest. To address this, researchers have spent efforts in the development of exogenous PA contrast agents, which are summarized in Table 1. Since exogenous contrast agents can be designed to have different absorption spectrum with significant red-shift, it could overcome the background imposed by intrinsic chromophores and allow deep imaging depth, providing visualization of cellular and molecular events beyond the anatomical and functional information with high specificity.

### 3.1. Principle of Exogenous Contrast Agent Design

An ideal exogenous contrast agent has to be target-specific, and yet be able to generate signals that are distinguishable from the intrinsic contrast. In general, exogenous contrast agents consist of two major components, the signal generating compound, and the targeting ligand [69]. Depending on the chemical nature and stability of both components, different assembly strategies are employed. Direct conjugation of the targeting ligand to the PA compound is the most straightforward method, and is often used for gold nanostructures with the use of thiol-gold chemistry for surface modification [40]. Polymer encapsulation is often used for small molecular dyes, which often consist of huge and conjugated hydrocarbon systems that are hydrophobic; thus, insoluble in water. Therefore, a polymer capsule is added to grant water solubility and surface for functionalization [43,70,71].

#### 3.1.1. Targeting Mechanisms and Ligands

The targeting mechanisms can be categorized into active or passive targeting, which takes advantage of the structural differences between normal and tumorous tissues. Tumor tissues often develop vast and hyperpermeable vasculature system while lacking lymphatic drainage, leading to the enhanced permeability and retention (EPR) effect where exogenous contrast agents up to ~100 nm in diameter can be retained at the tissue without specific targeting ligand or adsorption mechanism [72]. However, passive targeting is not applicable to contrast agents with a diameter larger than 100 nm since they will be cleared via the reticuloendothelial system (RES) [73]. Passive targeting relies heavily on tissues’ structural properties such as dense vasculatures and the absence of lymphatic ducts, leading to the EPR effect. Therefore, passive targeting normally suffers from low labeling efficiency of the contrast agents due to the lack of specificity and confound targeting strategies. A high dosage of the contrast agents might be needed to compensate for the confound targeting strategies which might lead to a high cytotoxicity, further decreasing the biocompatibility of the contrast agents. Hence, active targeting is ideal for exogenous contrast agent design. The biological target for active targeting should either be overexpressed in the early development of the disease or has a low expression in off-target tissues. Although a wide range of molecules such as affibodies [74], aptamers [75], proteins, and polymers can be used to achieve active targeting, factors such as biodistribution, biocompatibility, cytotoxicity, and assembly strategies should be considered. For example, targeting ligand, such as 2-deoxy-D-glucose, is very small (<1 nm) in size, making it highly permeable through physiological barriers, and can be easily egested via renal clearance. On the contrary, large targeting ligands, such as monoclonal antibodies (>15 nm), have high specificity and affinity at the expense of biodistribution and clearance due to its bulkiness. However, affibodies, small (~6.5 kDa) affinity proteins, have been in the spotlight for primary tumor-targeting over the past five years owing to their high specificity, high affinity, and minimal immunogenicity [76]. Affibodies has been one of the most frequently used targeting ligands for cancer imaging since affibodies for human epidermal receptor 2 (HER2) and epidermal growth factor receptor (EGFR) are well-studied cancer biomarkers and very robust with pico-molar sensitivity [74].

#### 3.1.2. Photoacoustic Compounds for Exogenous Contrast Agents

The signal generating compound should have a high photostability in order to avoid irreversible chemical changes such as denaturation or photodegradation upon continuous excitation. In terms of the optical absorption spectrum, an ideal signal generating compound should have a peak absorption in the NIR region for deep-tissue imaging since NIR light has the optimal balance between optical scattering and absorption. Moreover, the absorption peak should be sharp so that the PA signals generated by the contrast agent can easily be distinguished from intrinsic background chromophores, allowing easy spectral unmixing even at low molar concentration and limited number of excitation wavelengths. Nevertheless, the compound should have a high molar extinction coefficient to maximize the amount of light absorbed, and a low quantum yield to increase non-radiative energy dissipation such that strong PA waves can be generated.

### 3.2. Nanoparticles

Nanoparticles (NPs) are a diverse group of contrast agents that are commonly used in PAT. This is because NPs offer a high degree of freedom for their material, shape, and size, giving rise to a myriad of NPs with a great variety of morphologies and surface modifications. However, bioaccumulation of NPs often occurs since they are typically 10–100 nm in size, which cannot be cleared via the RES [69]. Although not enough clinical evidence is available to indicate cytotoxicity over long-term accumulation, hepatic retention remains a concern for using NPs as PA contrast agents. In addition, the reproducibility and purification of homogenous NPs remain as challenging tasks.

#### 3.2.1. Metallic Nanoparticles

Among the great variety of NPs, gold NPs (AuNPs) had been the focus. AuNPs are one of the most promising NPs for exogenous PA contrast since gold possesses a tunable photophysical property as NPs, and the strong thiol-gold bonding allows covalent surface modifications for optimization in specificity and biocompatibility. AuNPs utilized a phenomenon called surface plasmon resonance (SPR) effect to generate PA signals. When an incident light illuminates the surface of an AuNP, the conduction electrons at the surface will oscillate, collectively forming localized plasmons. These plasmons will then decay and release heat energy that can be detected for PAT. The resonant frequency of the localized plasmons strongly depends on the geometry and size of the AuNPs. Therefore, the wavelength required for localized SPR (LSPR) to occur can be tuned accordingly [77,78]. Moreover, AuNPs, unlike many other fluorophores, are not susceptible to photobleaching, making it a robust and stable contrast agent. Gold nanospheres (AuNSs) are the most fundamental shape for AuNPs. AuNSs have a single absorption peak at 520 nm, which is similar to the absorption of hemoglobin, making the generated PA signals difficult to be distinguished. Attempts had been made to red-shift the absorption peak to approximately 600 nm by increasing the size up to 100 nm [79]. However, it would be ideal if the peak absorption falls into the NIR window where photons are least scattered and absorbed by biological tissues. By lumping AuNSs into a gold vesicle, the SPR effect is enhanced; thus, red-shifting the absorption peak into the range of 650–800 nm at the expense of broadening the absorption peak [80]. Since the discovery of LSPR depends on the AuNPs geometry, a myriad of gold nanostructures, including gold nanorods (AuNRs), gold nanocages (AuNCs), gold nanoshells (AuNShs), and gold nanostars (AuNSts) had been synthesized [40]. In addition to gold, the SPR effect had been exploited to create other noble metal NPs from silver (Ag) [49], platinum (Pt), and palladium (Pd) [81]. Studies had shown silver NPs (AgNPs) exhibit the sharpest and strongest band among other noble metals [78]. However, gold being inert and biochemically safe are chosen to optimize the biocompatibility. Nevertheless, the changes in optical absorption spectra of silver NPs in different morphologies (Figure 6) such as spheres, cubes, and shells shed light on the possible spectral tuning on its gold counterpart [82,83].

An increase in the number of edges or sharpness of the NPs will increase the charge separation, resulting in red shifting the peak absorption, whereas the higher-order symmetry will increase the LSPR intensity. Depending on the shape of the NPs, several modes for polarization give rise to their corresponding resonance absorption peaks. For instance, NRs have two polarization modes, giving two absorption peaks typically at 520 nm for transverse polarization, and longer wavelengths for longitudinal polarization [78,84].

AuNRs are the most extensively used exogenous PA contrast agents since the NRs have a facile synthesis route and tunable absorption in the NIR region by altering the aspect ratio. Chen et al. had synthesized miniature AuNRs (width: 8 nm; length: 49 nm) that has a peak absorption in the NIR-II region (1000–1700 nm) that has similar aspect ratio as regular AuNRs (width: 18 nm; length: 120 nm) at 5–11 times smaller in dimension [41]. The miniaturized AuNRs enabled 4.5 times greater PA signal generation compared to regular-sized AuNRs. Numerical calculations had shown a quadratic relation between the PA amplitude and the surface-to-volume ratio of each NR when optical absorption is matched across AuNRs with different sizes. Although the contrast agent retention in the tumor tissue is significantly higher than the regular AuNRs, similar bioaccumulation of the AuNRs is observed in the liver, kidney, and spleen due to the inability for reticuloendothelial clearance [41]. Kim et al. had demonstrated the use of AuNRs as an in vivo tracer for sentinel lymph node mapping, showcasing the enhanced imaging depth and contrast with AuNPs. AuNRs and AuNShs targeting endothelium inflammatory biomarker, ICAM-1, had been used for IVPA imaging in mouse vena cava for identifying the atherosclerotic plaque and locating the endothelial inflammation [39,42]. Nonetheless, polyethylene glycosylated (PEGylated) AuNShs and AuNCs had been used for imaging mouse brain vasculature after intravenous injection. Images of the brain vasculature show clearer and more detailed structures when compared to the one without injection, where blood vessels with 100 μm can clearly be seen with PEGylated hollow AuNShs. I. Sun et al. have showcased a gas-generating AuNP possible for enhanced contrast in both ultrasound and PAT [44]. The laser-triggered gas-generating AuNPs (AzGC-AuNPs) was developed by conjugating azide compounds such as 4-azidobenzoic acid (AzBA) onto the amine group of glycol-chitosan-coated AuNPs. Upon NIR illumination, the AuNP acts as a photocatalytic agent, catalyzing the photolysis of AzBA, and generating nitrogen gas which enhances the ultrasound contrast, showing potential to be used in PAT [44]. AuNPs had been a very promising exogenous contrast owing to the tunable SPR effect and the facile thiol-gold surface chemistry for biomolecules immobilization. However, more robust synthesis pathways are needed for the production of AuNPs except for AuNSs, AuNRs, and AuNCs, but more surface modifications such as silica coating are also needed to avoid plasmonic AuNPs changing their shapes as they are being heated to shape-transition temperature by the laser [85]. Long-term study for the toxicology of AuNPs bioaccumulation is needed before AuNPs can be used as a prominent contrast agent for clinical purposes [86].

Noble metals such as silver (Ag) had also been used to synthesize nanostructures as contrast agents for PAT. AgNPs, similar to AuNPs, exhibits LSPR phenomena with different morphologies depicted in Figure 6. Using Ag nanosystems has both advantages and disadvantages over Au nanosystems. One of the major advantages is that Ag is known to have better optical absorption properties, including ~10% stronger absorption and sharper plasmon resonance when compared to Au, thus providing a stronger PA signal [50,87]. The well-known antimicrobial property of AgNPs has also drawn attention in using Ag nanosystems for clinical PAT. However, one of the major disadvantages of Ag is its relatively high reactivity. Lee et al. have synthesized porous Ag-Au alloyed nanoparticles (pAgAuNPs) to utilized both the more superior optical properties of Ag, and the chemically inert property of Au [51]. Compared to AuNPs, the pAgAuNPs gave 2.7 times stronger PA signals for whole-body in vivo imaging experiments in mice. Different combinations of noble metals, such as Ag-Au [48], Pt-Ag, and Pd-Pt had been reported to be used to synthesize metallic double-shell hollow NCs [81]. Nonetheless, the higher reactivity of Ag over Au has led to concerns over its cytotoxicity which is highly dependent on local concentration within different biological tissues [49,50,87]. A more sophisticated and long-term study for the biodistribution and cytotoxicity of AgNPs is needed to determine its efficacy and biocompatibility.

Among all noble metals, Au, Ag, and Pt are the common metals used in nanomedicines for both cancer therapies and other biomedical applications such as contrast agents for PAT [88]. However, all three noble metals are relatively scarce; hence, their high cost for synthesis. One of the widely used noble metals for organic synthesis, Pd, has gained attention in the biomedical community for its similar optical properties to Au and Ag, including high thermal stability and relatively low cost in production [89]. Pd has a higher bulk melting point (MPPd = 1828K) compared to Ag (MPAg = 1235K) and Au (MPAu = 1337K), which should translate to better photothermal stability under laser illumination [53]. Huang et al. have reported ultrathin hexagonal Pd nanosheets that exhibit efficient photothermal conversion within the NIR region, and higher photostability compared to AgNPs and AuNPs [52,53]. However, the cytotoxicity of Pd nanostructures is reported to be cell-type specific and highly dosage-dependent. Size-dependent retention in spleen, liver, heart, and lung at variable duration is also reported [90]. Similar to other metallic nanosystems, more cautious and detailed understandings are needed before employing metallic NPs for clinical PAT.

In addition to elemental metallic NPs, inorganic dyes NPs such as Prussian blue nanoparticles (PBNPs) have been reported to be used to synthesize multimodal PA contrast agents. One of the advantages of utilizing Prussian blue is that it is an ancient dye that has been approved by the Food and Drug Administration (FDA) for human biosafety. Moreover, Prussian blue can be synthesized at a relatively low cost by mixing ferric cations with ferrocyanide solution. Fu et al. have demonstrated the use of PBNPs as a photothermal ablation agent [54]. Dumani et al. have developed a method for size-controlled synthesis of photomagnetic Prussian blue nanocubes (PBNCs) using superparamagnetic iron oxide nanoparticles (SPION) as the precursor [55]. The PBNCs are reported to have high magnetic saturation and an optical absorption peak at 700 nm. With superior photostability, PBNCs have been used as contrast agents for both MRI and PAT in vivo [55].

#### 3.2.2. Carbon Nanotubes

Another class of NPs is the graphene-based single-wall carbon nanotubes (SWNTs). SWNTs possess an intrinsic absorption in the NIR region, where spectral tuning can be achieved by changing their diameter and chirality [91]. Although SWNTs have a lower extinction coefficient compared to AuNPs, a variety of small molecules, drugs, and polymers can be linked to the surface of SWNTs for additional functionalities and biocompatibility optimization [92]. La Zerda et al. had demonstrated the use of arginylglycylaspartic acid, commonly referred to as the RGD peptide motif, linked SWNTs to achieve αvb3 integrin targeting for primary tumor and angiogenesis imaging with improved contrast and sensitivity in mice [56,57]. However, multiple studies have reported the potential of cytotoxicity and inflammatory response for graphene-based NPs. Further investigations on understanding the biocompatibility of SWNTs and ways for optimization are needed [93].

### 3.3. Molecular Photoacoustic Contrast Agents

Molecular PA contrast agents (MPACs) are of great interest to the community due to their diverse structure and design available for spectral tuning, and their inherent small molecular size optimized for permeability and retention in tumor vasculature. A desirable MPAC should possess a high ground state molar extinction coefficient within the NIR region for achieving deep imaging depth in biological tissues. It should also exhibit a linear PA response allowing a quantitative concentration projection. Moreover, the ideal MPACs should have high solubility and stability in vivo with minimal toxicity. The library of MPACs has been rapidly growing, and many different chemical structures have demonstrated PA responsive properties. It is difficult to categorize all MPACs into a few specific classes of molecules. However, some of the MPACs belonging to small and well recognizable families of established chromophores such as cyanine and tetrapyrroles will be covered in this review [71].

#### 3.3.1. Cyanine Dyes

Cyanine dyes have a fundamental structure made up of two nitrogen-containing moieties, typically indoline, oxazole heterocycles, linked by a linear polymethine chain which consists of odd number carbon atoms typically 3, 5, or 7. One of the nitrogen atoms in the heterocycles is usually positively charged and is resonant with its counterpart via the conjugated polymethine chain. About 100 nm bathochromic shift for the absorption and emission can be achieved by extending the polymethine conjugation system by one vinylene moiety [94,95]. Common indocyanine dyes such as Cy3(1), Cy5(2), and Cy7(3) (Figure 7) are often used in fluorescence imaging owing to their tunable absorption and emission spectra. One of the representative cyanine dyes used for PAT is the indocyanine green (ICG), which is initially a fluorescence dye used for in vivo ophthalmic angiography that is approved by the FDA due to its low toxicity. ICG is a Cy7 derivative where a butane sulfonic acid/sodium sulfonate sidechain is added to each nitrogen atom in the benzoindole heterocycles. Therefore, it is water-soluble and has an excitation peak at 780 nm.

Properties such as low toxicity, high water solubility, and high NIR absorptivity have made ICG and its derivatives a popular choice for PAT and medical diagnosis [38,96]. ICG has been used in fusion with different targeting affibodies to label cancer biomarkers such as HER2 [97] and B7-H3 [59] for early-cancer imaging with photodynamic therapy (PDT) capability. Strategies such as PEG conjugation and micellar formation have been employed to enhance the permeability and retention of ICG via the EPR effect. Shi et al. has demonstrated PAT guided chemo-photothermal therapy (PTT) by devising a bimodal drug delivery micelle using an amphiphilic copolymer made of three monomers, including (i) PEGylated isocyanide, (ii) ε-Caprolactone, and (iii) rhodamine B, which are self-assembled and loaded with IR780 (ICG derivative) and camptothecin (a chemotherapeutics) [98]. Furthermore, Mishra et al. have designed a PA responsive Ca^2+^ chemosensors based on IR780 appended with Ca^2+^ chelating moiety used in PAT for molecular sensing [99].

Although cyanine dyes have been widely used as MPACs owing to their low toxicity and NIR absorptivity, there are several drawbacks. The optical properties for ICG vary drastically in the aqueous environment due to aggregation and unspecific binding to biomolecules such as proteins, lipoproteins, and phospholipids [58]. Cyanine dyes are among the most comprehensively studied class of MPACs. For example, dyes based on squaraine [100,101], curcumin [71], BODIPY [102,103], and xanthene [63] structures have been synthesized and demonstrated their PA properties. However, the relatively blue-shifted absorption peaks compared to tetrapyrrole structures limited their biomedical applications. One notable xanthene dye, methylene blue (MB), has been widely used as a contrast agent in PAT. Song et al. had injected MB into a rat for noninvasive visualization of sentinel lymph nodes which otherwise, requires an invasive surgical procedure [104]. Wang et al. had demonstrated a 492-folds increase in PA signal with MB encapsulated in sodium dodecyl sulfate (SDS) micelles [61]. Recently, Jeevarathinam et al. utilized the redox chemistry of MB to photoacoustically quantify drug release from nanocarriers [62]. Paclitaxel conjugated to a chemically reduced form, leuco form methylene blue (LMB), which was encapsulated in a poly(lactic-co-glycolic acid) NP. Upon the release of paclitaxel, the PA silent LMB will be spontaneously oxidized into PA active MB. By monitoring the increase in PA signals generated by the MB, the amount of paclitaxel release can be quantified [62]. Nonetheless, most of these MPACs, like cyanine dyes, are susceptible to photoisomerization, photobleaching (Figure 8), and solvatochromism. The lack of photostability in cyanine dyes under NIR illumination has greatly limited its potential. Upon light absorption, besides fluorescence emission, some of the electrons in the first excited singlet state are shifted to the first excited triplet state via an intersystem crossing. The long-lived triplet state is then quenched by oxygen molecules, giving rise to reactive oxygen species such as singlet oxygens [105]. These highly reactive singlet oxygens will attack the polymethine chain, and subsequently, photodegrade the cyanine molecule via C–C bond cleavage [106]. These undesired phenomena will alter MPACs’ intrinsic absorption spectra, hindering quantitative PA measurements [107].

#### 3.3.2. Tetrapyrrole

Tetrapyrrole is a class of molecules composed of four pyrroles or pyrrole-like rings, creating a highly conjugated and planar macrocyclic core. Due to their planar geometry for the cyclic tetrapyrroles, J-aggregation mode has been used to enhance the PA properties, and induce a 30 nm bathochromic shift for the absorption peak [108]. Morales et al. utilized the novel J-aggregation of a phthalocyanine-based dye (*λ_max_* = 780 nm) to develop a DNA-based nanosensor for PA detection of interferon-gamma [65]. Tetrapyrrole structure has proven to be a good light harvester demonstrated by the chlorin present in chlorophyll molecules, which is an efficient photochemical center for photosynthesis [109]. Huang et al. utilized the metal coordinating properties of naphthalocyanine core to synthesize a PEGylated tin (IV) chloride octabutoxy metallonaphthalocyanine (*λ_max_* = 930 nm) for visualizing brain vessels in PAT [110]. Zhou et al. reported a dye with a similar structure (*λ_max_* = 997 nm) to demonstrate strong PA contrast up to a depth of 11.6 cm under chicken breast using a PACT system [111].

In addition to using the tetrapyrrole-based structure as an individual MPAC, both linear and cyclic tetrapyrroles can be used as a cofactor for apoproteins to create switchable PA contrast agents. Heme, the cofactor in hemoglobin, consists of a porphyrin ring (a tetrapyrrole structure) with Fe^2+^ ion coordinated by the center nitrogen atoms, which are responsible for the endogenous hemoglobin contrast [109]. Biliverdin Ixα, a linear tetrapyrrole and an enzymatic product of heme, is incorporated into the apo form of bacterial phytochrome (BPhP) to create a photo-switchable protein as a PA contrast agent. The photo-switchable capability is made possible with the photoisomerization of biliverdin upon photo-stimulation. Li et al. demonstrated in vivo detection of an engineered protein, which is composed of a photosensory core module (PCM) in bacterial phytochrome protein (BphP) from *Deinococcus radiodurans,* termed DrBphP-PCM. The protein was expressed by U87 cells in a mouse brain, which was imaged using a reversibly switchable single-impulse panoramic PACT (RS-SIP-PACT) system with a spatial resolution down to 125 µm. The PCM can be further segmented into three domains, PAS (Per-ARNT-Sim), GAF (cGMP phosphodiesterase/adenylate cyclase/FhlA), and PHY (phytochrome-specific). By truncating the DrBPhP-PCM into two sub-units (DrPAS and DrGAF-PHY), the first set of PA contrast agents, termed DrSplit, capable of detecting protein–protein interactions in deep-seated mouse tissues were reported, holding great potential for non-invasive deep-tissue PA functional imaging [67].

#### 3.3.3. Phototheranostic Molecular Photoacoustic Contrast Agents with Microcosmic Molecular Motion

One of the major challenges for MPACs is the lack of photothermal conversion efficiency, limiting their use in PAT and as a phototheranostic agent. This is because most of the MPACs utilize a donor–acceptor (D-A) coplanar structure, where strong intermolecular interaction significantly blocked other pathways of heat generation [112]. Liu et al. synthesized a series of NIR-absorbing dyes that exhibit PA response and photothermal efficiency better than that of AuNPs and ICG [68]. Such improvement is made possible by utilizing the process of twisted intramolecular charge transfer (TICT) [113,114], which upon photoexcitation, active molecular rotations promote non-radiative relaxation accompany with a bathochromic shift in emission. Unlike their previous work in aggregation-induced-emission which promotes fluorescence by restricting molecular motions, they introduced a triphenylamine (TPA), labeled orange in Figure 9A, as a molecular rotor and a secondary donor to the low bandgap and NIR-absorbing D-A-D core, labeled as brown and green in Figure 9A. Alkyl chains with various carbon atoms, depicted as R in Figure 9A and pink bubbles in Figure 9B,C, branch out in order to shield the TPA in the aggregation state, as shown in Figure 9B,C, promoting molecular rotation, hence increasing its photothermal conversion efficiency and PA response depicted in Figure 9D. Liu et al. are the first to utilize of TICT state to promote photothermal efficiency and demonstrated the use of a microcosmic molecular motion for PAT and biomedical applications, enabling new design strategies for MPACs [68].

## 4. Reflection-Mode Photoacoustic Microscopy

After going through different types of endogenous and exogenous contrast agents, it is essential to understand how the PA signals are detected in a PAT system. Among different PAT system configurations, a reflection-mode PAT system is the most suitable for a wider range of practical applications, including thick and excised tissue imaging and in vivo imaging. A single or an array-based piezoelectric ultrasonic transducer (UT) is commonly used to detect PA signals. Parameters such as the frequency, type of focus, and numerical aperture (NA) of the UT can be carefully chosen for different applications. In AR-PAM, a conical lens is typically used to create a ring-shaped incident beam that is focused on the sample. The UT mounted along the center of the ring-shaped beam is used to detect the generated PA signals [22,115]. Off-axis illumination has also been demonstrated in AR-PAM systems [116,117,118]. In PACT, the unfocused light beam is broadly illuminated on the sample with parallel acoustic detections. Different configurations, such as side and/or top illumination, have been demonstrated for different biomedical applications, e.g., finger joint imaging [119] and small-animal whole-body imaging [66,120,121]. For OR-PAM, in order to achieve high sensitivity and high spatial resolution, both the optical and acoustic foci are placed coaxially. Transmission-mode configuration is usually preferred for a simplistic and flexible design [122]. Despite its high sensitivity and high resolution, transmission-mode PAM’s potential for in vivo imaging is greatly limited by the thickness of the sample [25]. Therefore, a reflection-mode PAM system is a more practical design for a myriad of biomedical applications, especially in a clinical setting. However, one of the challenges for reflection-mode configuration is the incorporation of the opaque, and often bulky, UT along the illumination path without obscuring the light beam. In the following sections, we will first review representative components that had demonstrated the ability to circumvent such a challenge in reflection-mode PAM systems. In addition, three recent and promising all-optical detection methods will be reported, which can also be configured into reflection-mode PAM systems [123,124].

### 4.1. Optical-Acoustic Combiner

The most common configuration of a reflection-mode PAM system is to use optical-acoustic combiners, as shown in Figure 10a,b. In Figure 10a, two prisms are combined with a thin layer of silicone oil (SOL) [125]. The optical beam can pass through the SOL to excite tissues, and the generated PA waves will then be reflected by the SOL, which will be subsequently detected by a UT. To overcome the acoustic energy loss on SOL, an alternative design is to use a thin coating material (such as aluminum, ~100 nm in thickness) to reflect optical beam [126], while allowing the PA waves to pass through. However, the manufacturing difficulty is higher (Figure 10b). Optical-acoustic combiner allows a confocal and coaxial alignment of both the optical and acoustic beams, improving the sensitivity and resolution. To further improve the detection sensitivity, a concave acoustic lens, attached to the optical-acoustic combiner, can be used to focus the PA waves [127]. For instance, Li et al. developed a spherical concave lens with a high acoustic NA using a ray-tracing method [128]. The reported acoustic lens has a NA up to 0.74, compared with a conventional acoustic lens that has a NA of ~0.54, the sensitivity can be increased by 1.6-fold. Although the acoustic NA of the optical-acoustic combiner can be improved by optimizing the design of the acoustic lens, the optical NA is inherently limited by the long working distance of objective lens required to focus the light on the sample through the combiner, thus, limiting the lateral resolution of this type of PAM systems. In addtion, when the PA waves propagate from water into the combiner, over 30% of the acoustic energy is attenuated due to the acoustic impedance mismatch of the two media, resulting in weaker PA signal detection.

### 4.2. Ring-Shaped Ultrasonic Transducer

A customized ring-shaped UT (i.e., a hollow structure at the center), Figure 10c, can be incorporated into a reflection-mode PAM system [129]. The laser beam is focused on the sample through the ring-shaped UT, and the backward propagating PA waves would be detected by the same ring-shaped UT. Therefore, the objective and ring-shaped UT can be coaxially aligned to setup a reflection-mode PAM system. Moreover, due to a relatively high NA (~0.22) design of the ring-shaped UT, an objective lens with a high NA can be employed, enabling high-resolution imaging with simple system design. Wong et al. developed a UV-PAM system utilizing a ring-shaped UT for whole-organ imaging assisted by a microtome with a lateral resolution of 0.91 μm [36]. Furthermore, in this application, compared with an optical-acoustic combiner, the ring-shaped UT does not need to consider the UV absorption of the materials on the optical path.

### 4.3. Parabolic Mirror

As shown in Figure 10d, a parabolic mirror with a conical hole and a flat UT can also be implemented as a reflection-mode PAM [130,131]. The central conical hole allows the optical beam to focus on the sample directly. The excited PA waves would be focused and redirected by the parabolic mirror to the surface of the flat UT. Compared with the ring-shaped UT, the parabolic mirror-based detection method is more accessible because the flat UT is commercially available, and the parabolic mirror is easy to be customized and manufactured. Moreover, an objective lens with a high NA (~0.63) had been used to improve the imaging resolution [130].

For PAT with exogenous contrast agents, Li et al. developed a reflection-mode PAM system with an optical-acoustic combiner to track micro-rocket robots which were coated with 100-nm-thick gold layer, achieving single micro-robot in vivo imaging in the bloodstream of a mouse ear with a resolution of 3.2 µm under 532-nm pulsed laser excitation (Figure 11) [132]. Yao et al. demonstrated a super-resolution PAM system by imaging switchable bacterial phytochrome, achieving a high resolution of ~141 nm [66].

## 5. All-Optical Detection Reflection-Mode Photoacoustic Microscopy

Although the piezoelectric UTs are still the most commonly used detectors in PAT due to a wide variety of configurations and stable performance, all-optical ultrasonic detection methods have gained increasing attention for PAT in order to overcome the inherent bulkiness and opacity of piezoelectric UTs. The all-optical ultrasonic detection method can also improve the detection sensitivity and bandwidth, further improving the signal-to-noise ratio (SNR) and axial resolution. Here, we will review three major all-optical detection methods that have been employed in reflection-mode PAM, including microring resonator (MRR), remote sensing, and planar Fabry–Perot (FP) sensor.

### 5.1. Optical Microring Resonator

An MRR can be used as an ultrasound sensor, which contains a straight bus waveguide and a closed-loop waveguide (Figure 12a), forming a destructive interference at its resonance condition. When there is a pressure gradient due to the PA waves propagation, the ring-shaped waveguide would be slightly deformed, and the refractive index of the waveguide would be modified via the elasto-optic effect [133], thus shifting the resonance condition. The output intensity, which is modulated by the PA waves, is collected by a high-speed photodetector instead. Therefore, the PA signal is transferred into a detectable optical intensity.

MRR sensors, which are usually fabricated on a silicon substrate, have been investigated extensively. However, they are optically opaque [133,134,135], and thus an MRR-based PAM system can only be implemented in a transmission-mode configuration, limiting its applications to thin samples only. To develop an optically transparent MRR sensor for reflection-mode PAM, Li et al. fabricated a polymeric (SU-8) optical MRR on a fused quartz coverslip with an electron beam lithography system [136] (Figure 12a,b). The diameter of the ring-shaped waveguide is 60 µm, and the thickness of the cross-section is 800 nm. The gap between the ring waveguide and bus waveguide was optimized to be ~150 nm. The SU-8 MRR has a quality factor (Q-factor) of 10,400. As an ultrasonic sensor, it has a low noise-equivalent-pressure (NEP) of 6.8 Pa and a wide detection bandwidth up to 140 MHz (3-dB). With the wide detection bandwidth, they experimentally showed that the MRR sensor could provide an axial resolution of 5.3 µm by imaging a carbon-black thin-film pattern. The lateral resolution is 2.0 µm, with a 0.25 NA objective lens under 532 nm excitation. With the advantage of the sub-millimeter size of MRR, the same research group integrated the MRR into an endoscope, achieving a probe size of 4.5 mm in diameter [137]. In 2019, Li et al. integrated MRR into a chronic cranial window (CCW) with PAT for a longitudinal in vivo study [138]. The soft nanoimprint lithography fabrication allows cheap MRR production. Moreover, the stability and robustness of MRR in physiological environments, such as the bloodstream, can be achieved by applying polydimethylsiloxane (PDMS) coating on the MRR waveguide. The PDMS coated MRR is reported to maintain a stable and high Q-factors over 28 days. For the long-term in vivo cortical PAM imaging, MRR-based CCW successfully demonstrated the evolution and development of cortical vasculature (Figure 12c). MRR-based PAM system can resolve hidden vessels beneath the hemorrhage area (Figure 12d), which is unattainable in conventional PAM imaging, owing to the high axial resolution enabled by MRR. Therefore, MRR-based PAM system can provide abundant information for in vivo study.

### 5.2. Photoacoustic Remote Sensing

Remote sensing techniques, such as Michelson interferometer [139] and Mach–Zehnder interferometer [140,141], have been developed into PAT to avoid ultrasound coupling medium (i.e., water or ultrasound gels) that is inevitable in conventional PAM system. The interferometry-based sensors measure the PA signals by detecting the vibration of the sample surface induced by the PA pressure. Although these sensors are highly sensitive, they are also sensitive to the unwanted environmental noise, which greatly limits their applications in practical settings. Recently, a fiber interferometer based on a 3 × 3 optical coupler with optical quadrature detection has been proposed to detect nanometer-scale displacement without being affected by the environmental variation [142]. Further investigations are needed for in vivo biological imaging.

Instead of detecting the surface displacement, Hajireza et al. developed a non-interferometric method to measure the initial pressures before wave propagation [143]. A probe beam (1310 nm wavelength) alongside the excitation beam (532 nm wavelength) are coaxially focused on the sample, and the reflectance intensity is detected by a photodiode as a PA remote sensing (PARS) signal (Figure 13a). The pressure induced by the excitation laser beam would change the refractive index of the absorbing interface, thus modulating the intensity of the probe beam. Since the read-out signal is the reflectance intensity of the probe beam, PARS is insensitive to the unpredictable environment. PARS, in many ways, is different from conventional PAT systems or all-optical detection PAT systems. First, PARS directly detects the initial pressure induced by the excitation beam instead of the surface vibration. Second, the time-of-flight PARS signal cannot provide depth information. Third, the optical depth-of-focus determines the axial resolution. A SNR up to 40 dB has been reported in in vivo PARS imaging of mouse ear using ~40 nJ excitation pulse energy and 4 mW interrogation power (Figure 13b). To attain such a SNR, ~80 nJ pulse energy is typically required in a conventional PAM system [125]. With an objective lens of 0.4 NA, the PARS system reported a lateral resolution of 2.7 µm, and an axial resolution of 43.3 µm. To further extend the applications of the PARS system, they developed deep PARS (dPARS) microscopy, which can achieve high resolution beyond the optical transport mean free path of the excitation wavelength [144]. The excitation beam was loosely focused on the sample to generate PA signals beyond the optical transport mean free path, whereas the probe beam was tightly focused on the tissue. Therefore, the imaging depth was determined by the focused probe beam, which has a longer optical transport mean free path than that of the excitation beam, and the resolution of the dPARS system was optically determined by the focal spot size of the probe beam. They demonstrated carbon network imaging in intralipid scattering solution, achieving 2.5 mm imaging depth with an imaging resolution of ~7 µm. In mouse-ear imaging, they achieved 1.2 mm imaging depth with a SNR of ~50 dB using ~60 nJ excitation pulse energy.

### 5.3. Fabry–Perot Interferometer

A backward-mode FP interferometer (FPI) has also been developed to achieve high-resolution all-optical PAT, (Figure 14a) [145,146,147]. The FPI sensor head has a planar polymer (Parylene C) film spacer, which is sandwiched by two dichroic mirrors. The dichroic mirrors were designed to be optically transparent to the excitation beam (600–1200 nm) but highly reflective to the probe beam (other wavelengths). The excitation beam broadly illuminates the sample through the FP sensor, but the probe beam is focused and scanned across the FP sensor with the aid of a 2D galvanometer scanner. The reflectance of the probe beam introduced by the two dichroic mirrors at each scanning point is detected by a photodiode. When the thickness of the polymer is changed by the incident PA pressure, the reflectance intensity would be modulated, and thus the amplitude of PA signals can be detected. By using the FPI-based PAM system, researchers have demonstrated in vivo imaging for human palm vasculature [148,149], human peripheral limb arteries [150], mouse brain vasculature [146], and human placental vasculature [147]. Optical fiber-based FPIs have also been developed with high sensitivity, broadband, small footprint, and wide detection angels [151,152,153]. Chen et al. developed a tunable fiber-based FPI into PA mesoscopy, which can tune the resonant wavelength by using an additional 650 nm heating laser in real-time, achieving a high detection bandwidth of 30 MHz and a low NEP of 40 mPa/Hz^1/2^ [153]. Similarly, with the advantage of small size, Ansari et al. promoted fiber-based FPI into a PA endoscopy probe with an outer diameter of 3.2 mm [152]. They successfully demonstrated imaged the microvasculature in mouse skin with lateral and axial resolutions of 39 µm and 31 µm, respectively. For exogenous contrast agent imaging, Märk et al. developed their FPI-based PAM system with two excitation wavelengths to image a photoswitchable reporter protein (AGP1), achieving a differential image with high 3D resolution and low noise-equivalent concentration (3.6 µM) (Figure 14) [154].

## 6. Conclusions

In summary, PAT is a highly promising imaging modality suitable for a wide range of biomedical applications. Label-free PAT employs endogenous biomolecules such as hemoglobin, DNA/RNA, lipid, melanin, and water as the PA signal generating compound, providing non-invasive means in visualizing anatomical features for clinical applications such as cancer prognosis, and hemodynamic information for functional imaging with high spatiotemporal resolution. To unleash the full potential of PAT, active targeting exogenous contrast agents are needed to enable specific and disease-targeting imaging. This additional capability is essential for monitoring disease progression, evaluating therapy efficiency, and detecting a myriad of cancers at different stages. With future development, exogenous contrast agents can also be used as therapeutic agents for PTT/PDT. Nevertheless, owing to PAT’s ability to utilize different wavelengths of light, multiple contrasts can be imaged simultaneously, giving synergies between the endogenous and exogenous contrasts. Although exogenous contrast agents such as AuNPs and MPACs had proven to be promising PA contrast agents for deep-tissue imaging, biocompatibility in terms of water solubility, cytotoxicity, and renal clearance remains as a significant hurdle in translating both the exogenous contrast agents and PAT systems to the clinical setting.

Upon pulsed-light excitation, PA waves are generated due to thermoelastic expansion, which can then be measured by different techniques. A balance between the ease of construction and high performance is needed since a commonly used piezoelectric UT often obscures the optical path, restraining the design and configuration available for PAT systems. The implementation difficulty indirectly slows down the translation of PAT as a clinical tool for diagnosis. Several ways highlighted in the review have shown different approaches to circumvent such a limitation. However, more efforts are needed in optimizing the acoustic sensing process. The latest advances in all-optical acoustic sensing techniques removed the need of coupling medium for ultrasound transmission, and shed light on means that engineers can use to eliminate the inherent limitations in existing PAT systems. All these recent developments are paving the way to implement a miniaturized PAT system that utilizes both exogenous and endogenous contrasts, and providing synergistic advantages by combining the endeavors from research groups around the globe.

## Figures and Tables

**Figure 1 sensors-20-05595-f001:**
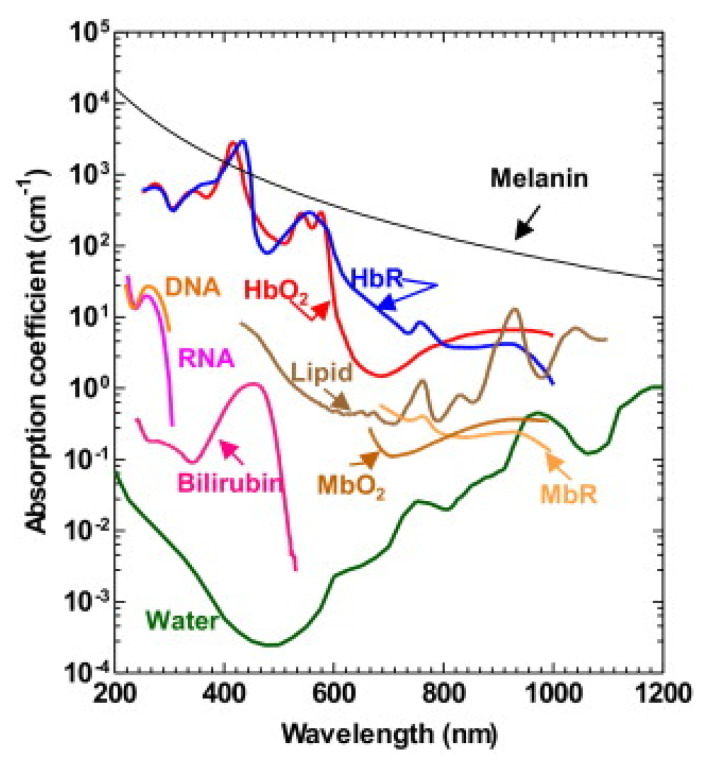
Absorption spectra of major endogenous contrast agents in biological tissue at normal concentrations. DNA and RNA, 1 g/L in cell nuclei; Oxy-hemoglobin (HbO_2_) and deoxy-hemoglobin (HbR), 150 g/L in blood; bilirubin, 12 mg/L in blood; lipid, 20% by volume in tissue; water, 80% by volume in tissue; melanin, 14.3 g/L in medium human skin; reduced myoglobin (MbR) and oxy-myoglobin (MbO_2_), 0.5% by mass in skeletal muscle. Reproduced with permission from Reference [9].

**Figure 2 sensors-20-05595-f002:**
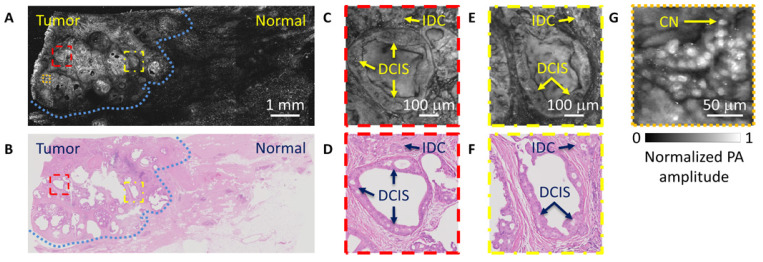
Comparison between label-free UV-photoacoustic microscopy (PAM) and H&E-stained images. (**A**) UV-PAM image of a fixed human breast tumor and (**B**) the corresponding H&E-stained histologic image. (**C**–**F**) Imaging of the same regions marked in (**A**) and (**B**), showing the similarity between UV-PAM and H&E-stained images. (**G**) Close-up image of the orange dashed region in (**A**) to show individual cell nuclei. IDC: invasive ductal carcinoma; DCIS: ductal carcinoma in situ; CN: cell nuclei. Reproduced with permission from Reference [25].

**Figure 3 sensors-20-05595-f003:**
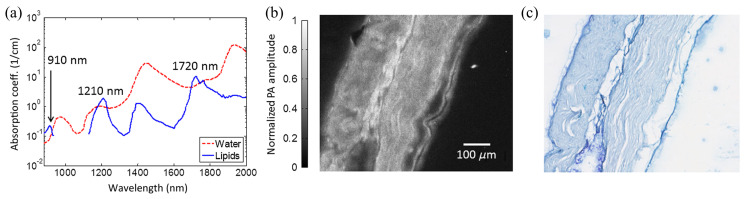
Label-free lipid imaging by PAM. (**a**) The absorption spectrum of lipids and water. (**b**) PAM image of a sectioned, unstained sciatic nerve, and (**c**) the corresponding stained image under bright-field optical microscopy. Reproduced with permission from Ref. [32].

**Figure 4 sensors-20-05595-f004:**
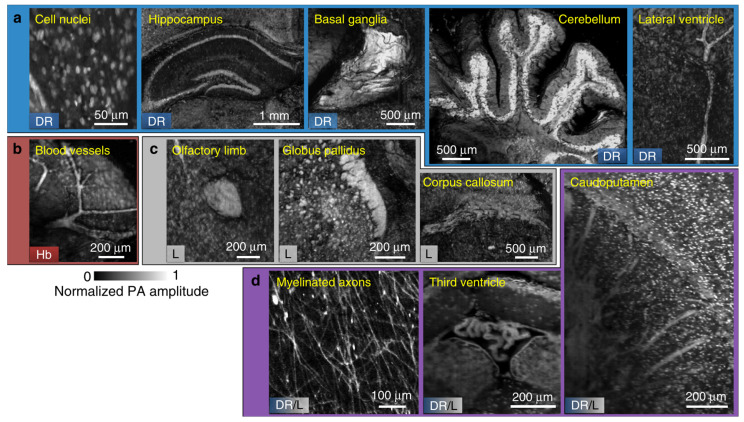
Image gallery of features extracted from label-free (microtomy-assisted) mPAM images of four unstained mouse brains embedded in agarose blocks. All features are shown in a coronal view. Collections of images showing the biomolecules that provide absorption contrast due to (**a**) DNA/RNA (DR), (**b**) hemoglobin (Hb), and (**c**) lipids (L). (**d**) Images of myelinated axons, third ventricle, and caudoputamen due to both DNA/RNA and lipids contrast. Reproduced with permission from [36].

**Figure 5 sensors-20-05595-f005:**
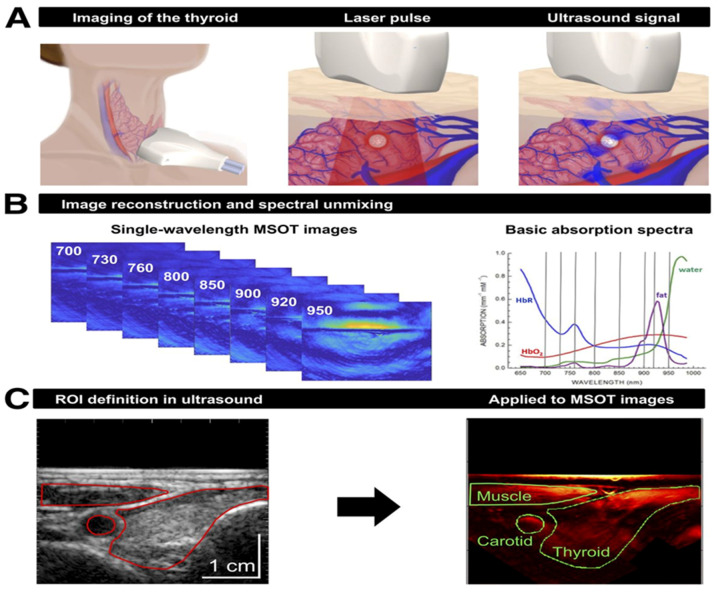
Principles of clinical multispectral optoacoustic tomography (MSOT) of thyroid. (**A**) Scheme of examination of thyroid gland with handheld hybrid MSOT/ultrasound system (left). Patients with thyroid nodules, healthy individuals, and Graves’ disease patients were scanned in a reproducible setup. Optoacoustic imaging is based on absorption of irradiated laser pulses within tissue (middle), followed by thermoelastic expansion and induction of ultrasound waves that can be detected with handheld detector (right). (**B**) In a first step, MSOT images are acquired for single wavelengths (left). Spectral unmixing, based on specific absorption spectra of different tissue constituents (right), allows assessment of functional parameters such as HbR, HbO_2_, fat content, and water content. (**C**) Transversal ultrasound image of thyroid gland and surrounding tissue allows exact localization of anatomic structures (left). ROIs drawn on ultrasound images were transferred to coregistered pseudo color-coded averaged MSOT images (here, HbT) for visual and quantitative analysis (right). Reproduced with permission from Reference [37].

**Figure 6 sensors-20-05595-f006:**
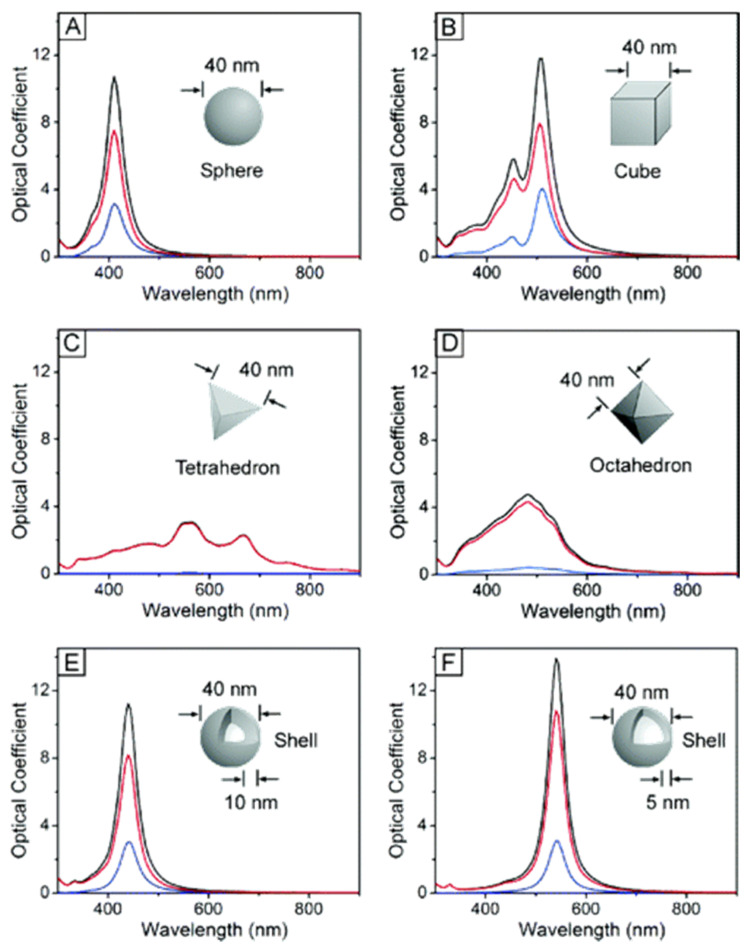
Calculated UV−vis extinction (black), absorption (red), and scattering (blue) spectra of silver nanostructures, illustrating the effect of a nanostructure’s shape on its spectral characteristics. An isotropic sphere (**A**) exhibit spectra with a single resonance peak. Anisotropic cubes (**B**), tetrahedra (**C**), and octahedra (**D**) exhibit spectra with multiple, red-shifted resonance peaks. The resonance frequency of a sphere red-shifts if it is made hollow (**E**), with further red-shift for thinner shell walls (**F**). Reproduced with permission from Ref. [83]. Copyright 2006 American Chemical Society.

**Figure 7 sensors-20-05595-f007:**
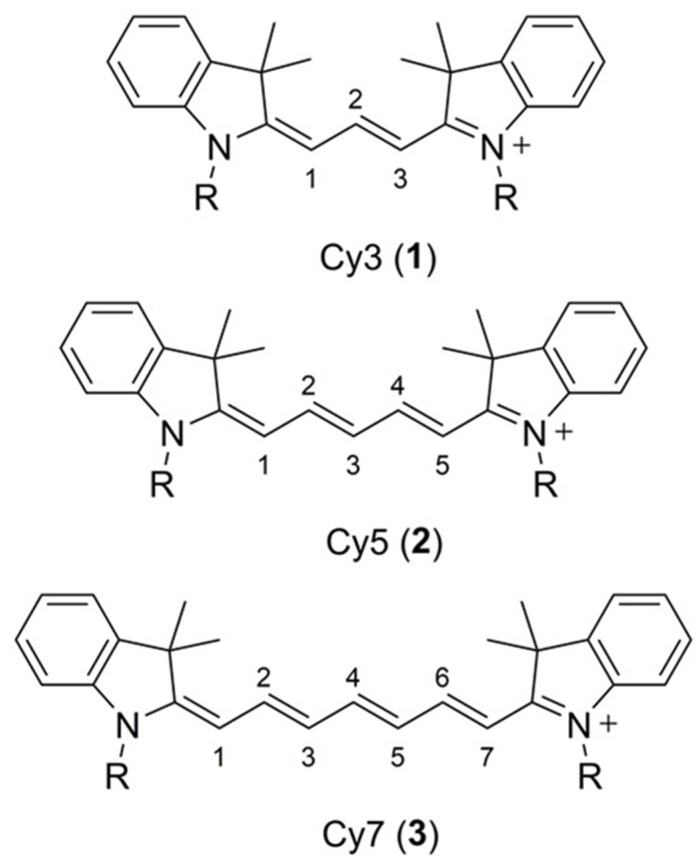
Molecular structures of the Cy3 (1), Cy5 (2), and Cy7 (3) indocarbocyanine based dyes. Reproduced with permission from Ref. [71].

**Figure 8 sensors-20-05595-f008:**
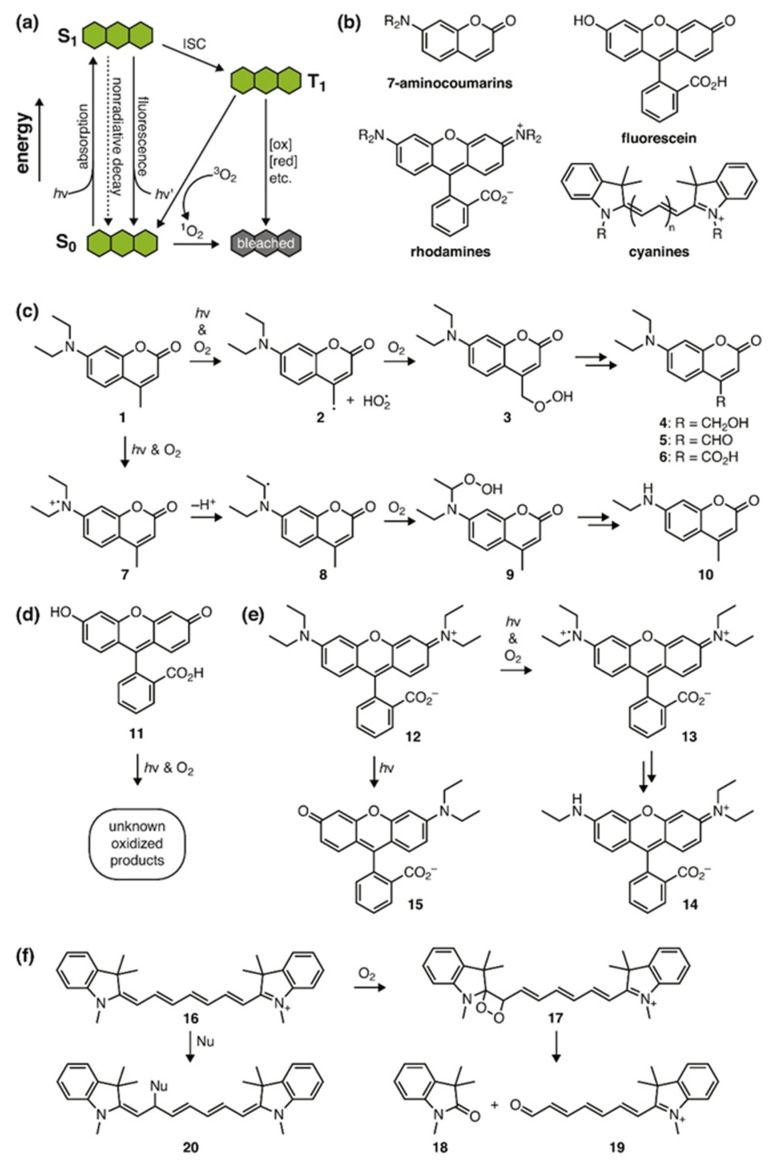
Common types of fluorophore photobleaching. (**a**) A general framework for the mechanisms of fluorophore photobleaching. (**b**) Generic structures of the major classes for small-molecule fluorescent probes. (**c**–**f**) Specific products and mechanisms for the photobleaching of (**c**) coumarins, (**d**) fluoresceins, (**e**) rhodamines, and (**f**) cyanines. Reproduced with permission from Reference [106].

**Figure 9 sensors-20-05595-f009:**
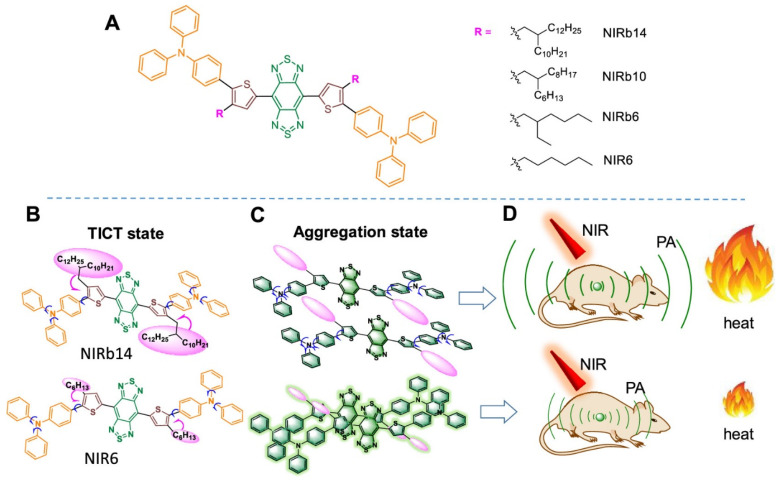
Molecular design of NIRb14, NIRb10, NIRb6, and NIR6 for photoacoustic (PA) imaging-guided photothermal therapy (PTT). (**A**) Chemical structure, (**B**) schematic illustration of the twisted intramolecular charge transfer (TICT) state in solution, (**C**) aggregation state, (**D**) scheme of the NIRb14, and NIR6 NPs for PA imaging-guided PTT. Reproduced with permission from Reference [68]. Copyright 2019 American Chemical Society.

**Figure 10 sensors-20-05595-f010:**
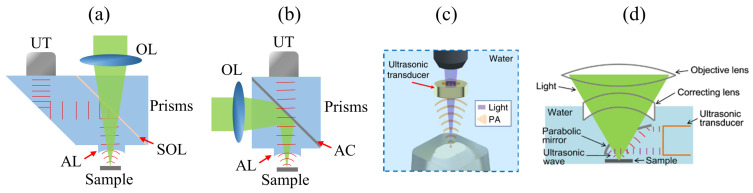
Reflection-mode PAM system configurations. Optical-acoustic combiner with (**a**) a SOL layer [125], and (**b**) a thin aluminum coating [126]. (**c**) Ring-shaped UT. Reproduced with permission from Reference [36]. (**d**) Parabolic mirror with a conical hole. Reproduced with permission from Reference [130]. UT: ultrasonic transducer; OL: objective lens; AL: acoustic lens; SOL: silicone oil; AC: aluminum coating; PA: Photoacoustic wave.

**Figure 11 sensors-20-05595-f011:**
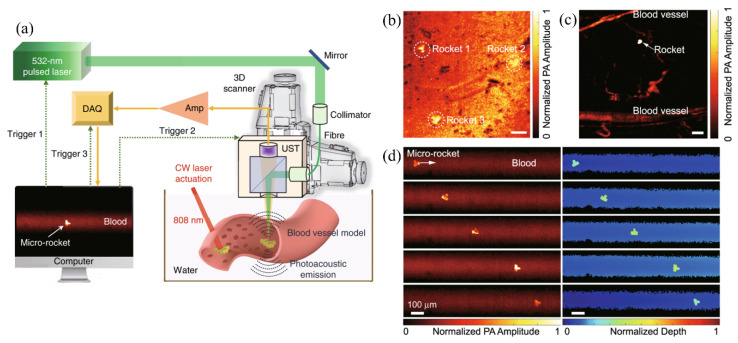
PAM of micro-robots in the bloodstream. (**a**) Schematic of the PAM system with optical-acoustic combiner for actuating and tracking micro-robots in the bloodstream. (**b**) PAM image of micro-robots in 500-µm-thick bovine blood. Scale bar: 100 µm. (**c**) PAM in vivo imaging of a micro-robot in the mouse ear. Scale bar: 100 µm. (**d**) Single micro-robot tracking using PAM in the blood vessel model. Reproduced with permission from Reference [132].

**Figure 12 sensors-20-05595-f012:**
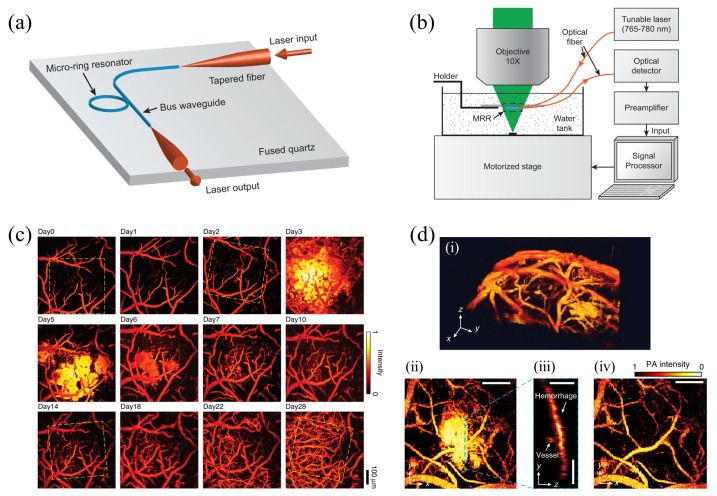
Applying the custom-made microring resonator (MRR)-based ultrasound sensor in PAM. (**a**) The structure of the MRR consists of a bus waveguide and a ring-shaped waveguide. (**b**) A representation system of MRR-based reflection-mode PAM. (**c**) Long-term PAM images of cortical vasculature in the same area over 28-day-period. (**d**) (i) 3D PAM image of a mouse brain contained a hemorrhage region. (ii) A maximum amplitude projected PAM image of the hemorrhage region. (iii) A PAM B-scan image of hidden vessels beneath the marked position in hemorrhage area in (ii). (iv) Vessels visualization beneath the hemorrhage layer. Scale bar, (ii–iv) 200 µm. Reproduced with permission from References [136,138].

**Figure 13 sensors-20-05595-f013:**
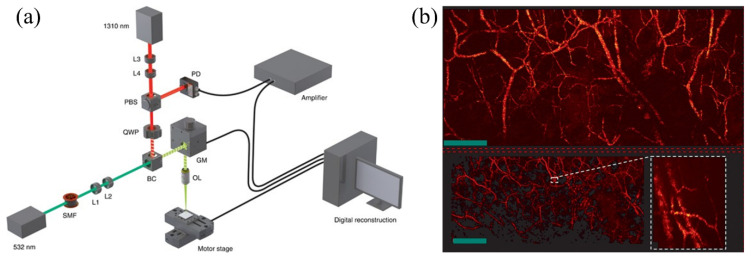
Vasculature imaging by PA remote sensing (PARS) microscopy. (**a**) System schematic of the PARS microscope. (**b**) PARS in vivo images of a mouse ear. Reproduced with permission from Reference [143].

**Figure 14 sensors-20-05595-f014:**
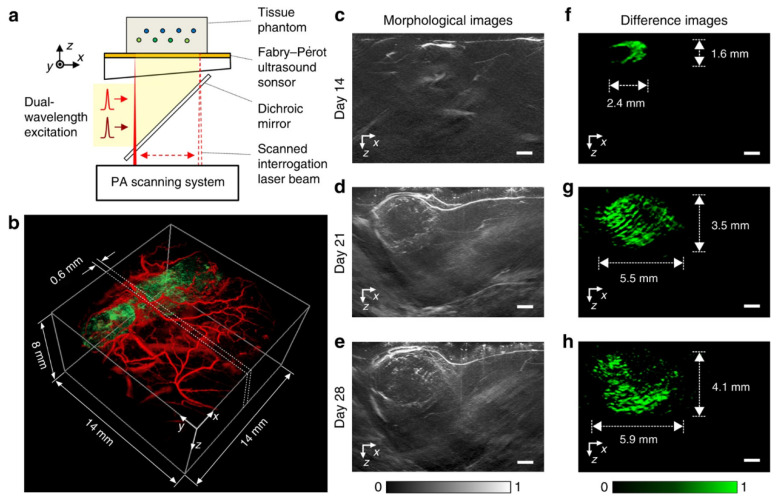
Longitudinal Fabry–Perot interferometer FPI-based PAM imaging of a mouse subcutaneous tumor of AGP1-expressing HT29 cells using dual-wavelength excitation. (**a**) System schematic of the PAM system with the FPI ultrasound sensor. (**b**) 3D PAM image of the tumor (green) surrounded by the vasculature in the skin and the underlying muscle tissue (red) acquired on day 21 post inoculation. (**c**–**e**) Cross-sectioned images acquired on (**c**) day 14, (**d**) day 21, and (**e**) day 28 post inoculation. (**f**–**h**) Difference images showing the distribution of AGP1-expressing tumor cells. Scale bar: 1 mm. Adapted with permission from Reference [154].

**Table 1 sensors-20-05595-t001:** List of exogenous contrast agents for Photoacoustic tomography (PAT).

Name	Size(s)	Preferred Wavelength(s) for Imaging	Demonstrated Biomedical Application(s)	Ref.
**Inorganic nanoparticles**				
**Gold nanorod**	Aspect ratio: 2	700 nm	Imaging of inflammatory response	[39,40]
**Miniature gold nanorod**	Width: 8 nm Length: 49 nm	1000–1700 nm	Molecular imaging of tumor targeting via gastrin-releasing peptide receptor	[41]
**Gold nanosphere**	20 nm	700–780 nm	Intravascular imaging of macrophages	[42,43]
**Gas-generating gold nanosphere**	17.45–35.72 nm	532 nm	In vitro contrast-enhanced ultrasound imaging	[44]
**Gold nanoshell**	Diameter: ~30 to 400 nm Thickness: 10–12 nm	800 nm	Photothermal ablation therapy; imaging of vasculature in rat brains	[45,46]
**Gold nanocage**	~46 nm	778 nm	Imaging of melanoma	[47]
**Gold/Silver hybrid nanorods**	Width: 12–14 nm Length: 50–55 nm Thickness: 10 nm	530, 800 nm	Monitoring silver release for antibacterial treatment	[48]
**Silver nanoplate**	Edge, Thickness: 25.3 nm, 10.4 nm; 60.9 nm, 12.5 nm; 128.0 nm, 18.0 nm; 218.6 nm, 25.6 nm	550, 720, 900, 1080 nm	Imaging of an orthotopic pancreatic tumor	[49]
**Silver-coated nanosphere**	180–520 nm	800 nm	Imaging of ex vivo canine pancreas	[50]
**Porous silver-gold alloyed nanoparticles**	50 nm	720–850 nm	Whole-body in vivo mouse imaging	[51]
**Porous flower-shaped palladium nanoparticles**	25–150 nm	808 nm	In vitro imaging of MBA-MD-231 cancer cells; In vitro photothermal therapy (PTT) for MBA-MD-231 cancer cell	[52]
**Ultrathin hexagonal palladium nanosheets**	28, 46, and 60 nm	826–1068 nm	In vitro PTT on hepatoma cells	[53]
**Prussian blue nanocubes (PBNCs)**	Edge length: Small: 20 nm Medium: 40 nm Large: 170 nm	700 nm	In vivo imaging of mouse abdomen upon subcutaneous injection of PBNCs in Matrigel	[54,55]
**Single-walled carbon nanotubes**	Diameter: 1–2 nm Length: 50–300 nm	780 nm	In vitro imaging of single-walled carbon nanotubes-indocyanine green-arginylglycylaspartic acid with U87 cancer cells	[56,57]
**Small organic contrast agents**				
**Indocyanine green**	<2 nm	810 nm	In vivo imaging in mice	[38,58,59,60]
**Methylene Blue**	<2 nm	650–700 nm	In vivo imaging for sentinel lymph node; Monitoring of drug release	[61,62]
**Evans Blue**	<2 nm	610 nm	In vivo capillary imaging; in vivo mouse brain imaging	[63,64]
**Phthalocyanine**	<2 nm	759 nm	DNA-based nanosensor for interferon Gamma detection	[65]
**DrBphP-PCM**		630, 780 nm	In vivo imaging in mice	[66,67]
**NIRb14 Dye**	<2 nm	800 nm	In vivo imaging for 4T1 tumor-bearing mice; PTT for 4T1 tumor-bearing mice	[68]

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
