# Peer review of "A Review of Endogenous and Exogenous Contrast Agents Used in Photoacoustic Tomography with Different Sensing Configurations"

_sensors, 2020, doi:10.3390/s20195595_

Round 1

Reviewer 1 Report

Dear authors,

The review article entitled “A review of endogenous and exogenous contrast agents used in photoacoustic tomography with different sensing configurations” written by Tsang et al. reports applications of endogenous and exogenous contrast agents for photoacoustic imaging with explanation of various systems. This review article can be accepted for publication on Sensors after the following minor issues on being addressed.

This review article generally described various applications of contrast agent with sufficient descriptions in the manuscript. To better explain and improve the readability of the manuscript, the reviewer recommend to the authors to include additional figures support the descriptions of exogenous contrast agent part (section 3). In addition, the authors also should modify the misprint and inadequate expressions such as overlapped part of Figure 5 and line 303 and word font style of description of Figure 4-6.

Author Response

Dear Editor and Reviewers,

Thank you for the care and time you took in reviewing our paper. Revisions have been made according to the valuable comments of the reviewers. Please find our responses in the attachment.

Best regards,

Victor Tsang

Reviewer 2 Report

The authors have attempted to review endogenous and exogenous contrast agents for photoacoustic tomography. While the review has accounted for some significant known and emerging concepts in photoacoustic contrast agents.

  1. The abstract section must include endogeneous and exogenous contrast agents – as it is the central theme of the review
  2. In the Introduction section
    1. Please introduce the concept of photoacoustics to the readers – It can help the readers from wide background
    2. It is necessary to highlight that the depth of detection in living tissue is enhanced by diffused photon and the readout signal being acoustic vibration.
  3. In line 49 Page 2, the statement that “ The molecules will undergo thermoelastic expansion” might be misleading – The mechanism must be explained as steps – 1. Excitation, 2. Conversion to heat, 3. Increased temperature around loci 4. Drop in temperature when irradiation was stopped.
  4. Please highlight the importance of first and second NIR windows – As it is relevant for the in vivo Imaging.
  5. In page 2, Section 2.1 Hemoglobin, Please discuss the absorbance spectral variation of oxyhemoglobin and deoxyhemoglobin.
  6. In section 2.3 Melanine, Please include the synthetic melanin nanoparticles and their use as photoacoustic contrast agent.
  7. In the nanoparticle section- Please include some innovative approach such as one described in - Nanoscale, 2019,11, 16235-16240.
  8. Under the cyanine dyes, - The photostability of the dyes is an important point to discuss.
  9. A focus on aggregation induced emission and design strategy to convert the phenomenon to favor photoacoustic imaging is appreciated! – I suggest a diagram to explain the concept – since it is emerging concept, it will be very helpful
  10. The article discusses the small section for targeting strategies, Likewise a section to include miscellaneous dye based or nanoparticle-based contrast agents can be discussed. Some example are -
    1. Chem. Int. Ed.202059, 4678. (redox dye indicate the drug release in vivo)
    2. Photoacoustics 2020, 18, 100166. (Prussian blue used for imaging)
    3. Some other materials include, Silver nanoparticles, Silver coated nanoparticles, methylene blue (lymph node imaging and drug monitoring), and Prussian blue
  11. It is suggested to expand the molecular and nanoparticle based contrast agent discussion at the expense of section 5 on All optical detection and reflection mode PAM. – This will help keep the focus of the discussion on the contrast agents than the techniques.

Author Response

(The authors gave the same response as above.)

Reviewer 3 Report

This manuscript is a review paper introducing various endogenous and exogenous methods in photoacoustic imaging, and reviewing the overall photoacoustic imaging (PAI) technologies with contact and noncontact methods. As a review paper it looks very informative and well-organized. It can be used as an introduction course leading to the PA world. However, it would be better if more recent technologies are introduced. Please pay more attention in order not to missing other good schemes and technologies! We hope our suggestions will improve the quality of the manuscript.

  1. Only the descriptions on the type of technologies are provided in 490~507. It would be better if the authors compare the advantages and disadvantages of the technologies also.

  1. At the section 5.2, only the Fabry-Perot scheme was introduced as an interferometer. It can be agreed that the Fabry-Perot is a good interferometer. However, it is recommended to introduce other technologies useful for the all optical sensing. It is thought that the following papers should be introduced; they utilized heterodyne optical interferometers for detecting acoustic waves.

-.“Noncontact photoacoustic imaging based on all-fiber heterodyne interferometer,” Optics Letters, 39, paper 16, pp. 4903-4906, 2014.

  1. In the Fabry-Perot interferometer scheme again, when the acoustic-induced surface displacement is smaller than the wavelength of the probing laser, the sensitivity of the interferometer becomes to depend heavily on the environment. The working point of the sensing system is floating! To overcome this stability problem many technologies have been reported. One of them is utilizing the optical quadrature (IQ) detection scheme. It is recommended to introduce the following paper and some of its reference papers. It used the fiber interferometer based on a 3X3 optical coupler. By utilizing the intrinsic phase difference among the output ports of the multi-port interferometer, it could detect down to a nanometer scale displacement without being affected by the environmental variation.

-. “Noncontact Photoacoustic Imaging Based on Optical Quadrature Detection with a Multiport Interferometer,” Optics Letter, Vol. 44, No. 10, pp. 2590-2593, 2020.

  1. It is a well-organized review paper. So many methods and skills are explained and compared. However, to increase the visibility of the paper it is recommended to make the table that can summarize the manuscript.

Author Response

(The authors gave the same response as above.)

Reviewer 4 Report

There are several reviews on the topic, so the immediate value of the manuscript is difficult to judge. However, the article can be improved in certain ways to make it valuable to the community. 

1) Provide a tabular breakup of contrast agents (CA) and classify them by manufacturing processes / clinical applications. 

2) Provide details of bio-compatibility and clinical acceptances / approved use cases. Given the authors talk about nano-particles, its an important dialogue to have.

3) The reason for putting the details of different PA configurations is not clear, it somehow looking a misfit in current manuscripts. However, the authors may use the configurations to justify use of certain contrast agents as preferred agents for certain configuration, if its holds true.

4) The list of citations is not comprehensive given the vast expanse of the topic. More references should be added and/or the topical focus of the article should be revised and sharpened. 

The general outlook and organization of the article needs to be worked on and improved to make the review useful.  

Author Response

(The authors gave the same response as above.)

Round 2

Reviewer 2 Report

The authors have addressed the concerns and given very reasonable explanation point by point. The manuscript is thoroughly revised to a satisfactory extent. The manuscript may be published as is in Sensors. 

Reviewer 4 Report

Changes are sufficient.